# REINFORCEMENT LEARNING WITH FINE-GRAINED REWARD FOR CONTROLLABLE TEXT GENERATION

## ABSTRACT

To alleviate text degeneration of large-scale language models and meet the requirements of real-world applications, it is essential to make generation more controllable. Previous reinforcement learning (RL) research on language modeling generally learns from sentence-level feedback, which requires extensive exploration to collect enough trajectories, and more steps to learn contributory components from a noisy trajectory corpus. To tackle that, we propose a novel reinforcement learning algorithm with FIne-grained REward (FIRE). We derive an extensible fine-grained reward function and ease the trade-off between reward approximation and training stability. We present a theoretical connection between our approach and canonical policy-gradient RL methods. Experimental results show that FIRE can achieve superior controllability of language models with less computational overheads compared to prior RL approaches.

## 1 INTRODUCTION

Large autoregressive language models (LLMs) trained on extensive corpus can generate high-quality texts. However, to satisfy real-world applications, making the generation more controllable is urgent. It is desired to reduce intrinsic defects of pretrained language models (e.g. toxicity, repetition) (Rae et al., 2021; Weidinger et al., 2021), and enhance specific attributes of generated texts for practical needs (e.g. positive sentiment for psychological escort, formality for academic writing) (Beltagy et al., 2019; Gu et al., 2022; Gururangan et al., 2020b). However, the deficient interpretability (Linardatos et al., 2021) of deep neural networks makes it challenging to guarantee the controllability of language models.

It is natural to retrain neural language models on domain-specific data (Keskar et al., 2019; Chan et al., 2021). However, since the parameter scales of large language models keep increasing, retraining is subject to computational overheads. Some researchers focus on post-processing methods (Yang & Klein, 2021; Liu et al., 2021; Krause et al., 2021), which control the generation by manipulating possibility distributions generated with a fixed LLM. They generally draw support from a small-scale attribute discriminator to regulate the possibility distribution for decoding. Hence these methods can hardly capture higher-dimensional features, which results in their limited controllability. Some researchers finetune the language models with partial parameters (Zhang & Song, 2022; Yang et al., 2023; Qian et al., 2022a), usually with continuous-prompt techniques (Li & Liang, 2021). However, they generally require additional domain-specific corpus. Moreover, with a supervised training schema, models are easily overfitted to the unwanted aspects beyond the required attribute and suffer from the discrepancy between training and inferring known as exposure bias (Schmidt, 2019).

Training language models on self-generated sentences can alleviate the above problems, which suits the reinforcement learning (RL) paradigm. RL-based methods (Lu et al., 2022a; Khalifa et al., 2021; Tambwekar et al., 2019; Guo et al., 2022) generally update language models with rewards, often designated as scalar heuristic metrics. However, RL feedback in NLP scenarios is generally sentence-level (or paragraph-level), since only after generating a complete sentence/paragraph can we score the text in previous settings. To control text generations towards specific attributes, this coarse-grained reward cannot provide clear guidance, since semantics vary while the sentence continues, often with twists or progression. Meanwhile, a sentence often contains massive functional components for syntax. Therefore, RL methods with coarse-grained feedback require more learning

steps and a larger exploration scale. It leads us to ponder whether we can propose finer-grained feedback to control the generation. However, how to propose a reasonable mechanism to discriminate the importance of different textual tokens and suffice it to be extended to diverse control requirements is non-trivial. Moreover, since the action space is substantially large (§2.1) in NLP scenarios, fine-grained control often requires value approximations to reduce computational overheads, which leads to a trade-off between value accuracy and training stability. This trade-off makes the RL training, which is known to be difficult to converge, even more unstable (§3.4).

In this paper, we introduce a novel reinforced learning algorithm with **FI**ne-grained **RE**ward named **FIRE**. First, we propose an extensible fine-grained reward function enlightened by a novel form of Bayesian factorization proposed in our paper. Second, to stabilize the training process, we transform the training objective to avoid involving specific reward values into the training objective. We also bridge a theoretical connection between our approach and canonical policy-gradient RL methods, which shows that FIRE is a more conservative version that updates parameters only towards high-confidence samples. We conduct experiments on 3 tasks: text generation with sentiment control, detoxification, and unlearning repetition. FIRE achieves on-par, usually better performance compared with competitive baselines. Notably, FIRE generally requires fewer learning steps to achieve superior performance compared with prior RL methods.

## 2 APPROACH

In this section, we first formulate the text generation process as a Markov Decision Process (MDP) in RL. Then we derive a new form of Bayesian factorization for controllable text generation, which enlightens us to propose a reward function of the token level. Finally, we describe our training objective to alleviate the trade-off between approximation and stability.

### 2.1 REINFORCEMENT LEARNING FORMULATION OF TEXT GENERATION

We first introduce the canonical undiscounted Markov Decision Process (MDP) in reinforced learning. A standard MDP can be denoted as $(\mathcal{S}, \mathcal{A}, \mathcal{T}, r)$. At each step, an action $a \in \mathcal{A}$ is made based on the current state $s \in \mathcal{S}$. Then the state will be transited to $s'$ with the possibility $\mathcal{T}(s'|s, a)$. A function $r : \mathcal{S} \times \mathcal{A} \to \mathbb{R}$ defines the returned reward based on the states and actions. The strategy is decided by a policy $\pi(\cdot|s)$, which is a predicted distribution over actions based on state $s$, which is trained to maximize the expectation of total rewards, known as action values:

$$Q(s_t, a_t) = \mathop{\mathbb{E}}_{\substack{a_{t+1} \sim \pi(\cdot|s_t) \\ s_{t+1} \sim \mathcal{T}(\cdot|s_t, a_t)}} \Big[ \sum_{t=1}^{H} r(s_t, a_t) | s_1 = s \Big] \tag{1}$$

where $H$ is the number of steps. Prior theoretical results show that the optimal policy $\pi^*$ satisfies the Bellman optimality equation:

$$Q^{\pi^*}(s, a) = \mathbb{E}_{a \sim \pi^*} \Big[ r(s, a) + \mathcal{T}(s'|s, a) \max_{a'} Q^{\pi^*}(s', a') \Big] \tag{2}$$

For text generation, the state can be defined as the partially generated sentence $y_{\leq i-1} = (y_1, y_2, \ldots, y_{i-1})$, and the action is the next token $y_i \in \mathcal{V}$ where the vocabulary $\mathcal{V}$ is the action space. The transition dynamic $\mathcal{T}(\cdot|s, a)$ is deterministic since each state-action pair $(y_{\leq i-1}, y_i)$ leads to a unique state $y_{\leq i}$.

In previous works (Lu et al., 2022a; Khalifa et al., 2021), rewards are generally returned after a whole sentence $y_{\leq L}$ is generated. They generally make the final feedback on behalf of the entire process and learn the entire trajectories of high-reward examples, which can be considered that action feedbacks from the same sentence are equal, formulated as $r(y_{\leq i-1}, y_i) = f(y_{\leq L}, c), i \in [1, L]$, where $f(y_{\leq L}, c)$ is a scorer, rating how well the current sentences $y_{\leq L}$ match the requirement $c$. This estimation limits the model performance and slows down the convergence speed as shown in our experiments §3.4.

### 2.2 FINE-GRAINED REWARD

To distinguish critical tokens from original sentences, we first reconsider the Bayesian factorization in controllable text generation, which is widely used in prior research (Yang & Klein, 2021; Krause

et al., 2021), and derive a new form as follows:

$$\mathcal{P}(y_i|y_{\leq i-1}, c_L) \propto \frac{\mathcal{P}(c_L|y_{\leq i})}{\mathcal{P}(c_L|y_{\leq i-1})}\mathcal{P}(y_i|y_{\leq i-1}) \tag{3}$$

where $c_L$ means satisfying the given attribute $c$ when the current sentence is extended to length $L$. The detailed derivation, differences compared to the canonical form and the application of prior Bayesian factorization in previous works can be seen in Appendix A. In Eq.3, $\frac{\mathcal{P}(c_L|y_{\leq i})}{\mathcal{P}(c_L|y_{\leq i-1})}$ is crucial for the next-token probability distribution. Even if $y_{\leq i}$ tends to highly satisfy the condition $c$ when the sentence extends to length $L$ i.e. $\mathcal{P}(c_L|y_{\leq i})$ is large, action $y_i$ may not play an important role since previous $y_{\leq i-1}$ may already make future generations satisfy the condition easily i.e. $\mathcal{P}(c_L|y_{\leq i-1})$ is large. It reveals that what really matters is the discrepancy between them. We deploy this intuition to the reward function, and propose the reward function as the logarithmic pattern of the crucial term in Eq.3,

$$r(y_{\leq i-1}, y_i) = \log \frac{\mathcal{P}(c_L|y_{\leq i})}{\mathcal{P}(c_L|y_{\leq i-1})}. \tag{4}$$

However, predicting $\log \mathcal{P}(c_L|y_{\leq i})$ with scorers generally deviates since traditional classifiers only provide available $\log \mathcal{P}(c_L|y_{\leq L})$. One solution is approximating each state through sampling (e.g. Monte Carlo methods) as follows,

$$\log \mathcal{P}(c_L|y_{\leq l}) = \mathbb{E}_{y \sim \pi(\cdot|y_{\leq l})}[f(y, c)] \tag{5}$$

where $f(y, c)$ is a pratical classifier rating how well a sentences $y$ match the attribute $c$. Unfortunately, the action space of language modeling is too large to enumerate abundant cases, which leads to a large deviation of the expectile. Therefore, we approximate the probability with subsequent $k$ tokens to elude computational overheads as follows,

$$\begin{aligned} \mathbb{E}_y[f(y_{\leq L}, c_L)] &= \sum_y f(y_{\leq L}, c) \prod_{i=l}^{|y|-1} \pi(y_{i+1}|y_{\leq i}) \\ &\approx \sum_y f(y_{\leq l+k}, c) \prod_{i=l}^{l+k-1} \pi(y_{i+1}|y_{\leq i}). \end{aligned} \tag{6}$$

Practically, we adopt nucleus sampling Holtzman et al. (2020) to sample $m$ cases for the expectile approximation. Current language models generally require an [EOS] token to finish generation. For this token, we use the sentence-level $-\log \mathcal{P}(c_L|y_{\leq L})$ as its reward since learning requires more samples of the desired attribute generated from the exploration.

## 2.3 TRAINING OBJECTIVE

In standard policy-based RL methods, given the trajectory $(s_1, a_1, r_1, s_2, a_2, r_2, \cdots)$, the training objective is to maximize the reward expectation, which is usually applied in the form of Eq.16. Considering approximation in calculating rewards and large action spaces of NLP scenarios, sticking with this training objective makes training unstable and hard to converge as shown in §3.4. Inspired by recent quantized reward conditioning schema (Lu et al., 2022a), we transform the training objective to avoid involvement of specific reward value. We sort and quantize reward values to pick up the highest/lowest $q$-quantile denoted as $r_h/r_l$. Then we convert our training objective to a form that maximizes the likelihood of tokens in the trajectory with rewards over $r_h$. The formula is as below:

$$\mathcal{J}(\theta) = \mathbb{E}\Big[\sum_n \log \pi(y_{n+1}|y_{\leq n}, \theta)\mathbb{1}(r_n > r_h)\Big], \tag{7}$$

where $\mathbb{1}(\cdot)$ is an indicator function to indicate whether the condition is satisfied. In the actual deployment, we also encourage the unlikelihood of tokens with rewards under the lowest quantile $r_l$ as follows:

$$\overline{\mathcal{J}}(\theta) = \mathbb{E}\Big[\sum_n \log \pi(y_{n+1}|y_{\leq n}, \theta)\mathbb{1}(r_n < r_l)\Big]. \tag{8}$$

Inspired by Proximal Policy Optimization (PPO), we add a KL-divergence penalty to loss to prevent the language model from deviating too far, which may destroy the original semantic space. We also

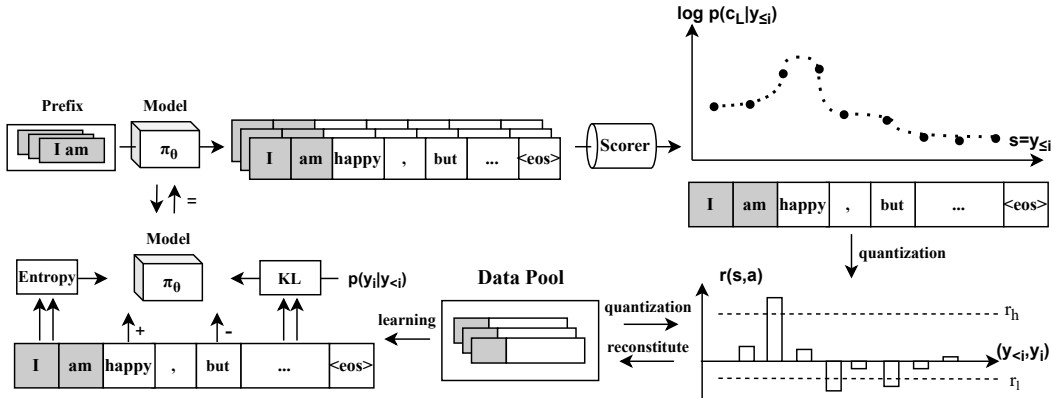

Figure 1: An overall framework of FIRE. During the exploration, the current policy model generates continuations of a given corpus. Then a corresponding scorer approximately calculates reward of each action. We then interpose data from exploration into the data pool and quantize all rewards to obtain $q$-quantiles. Then the policy model updates according to our training objective. The model circulates Exploration-Learning before the training ends.

add an entropy regulator to maintain the action diversity of the policy model. The training objective to maximize is supplemented as:

$$\widetilde{\mathcal{J}}(\theta) = \mathcal{J}(\theta) + \mathbb{E}\Big[\beta \sum_n \mathrm{Ent}\Big(\pi_\theta(y_{n+1}|y_{\leq n})\Big)$$
$$- \gamma \sum_n \mathrm{KL}\Big(p(\cdot|y_{\leq n})||\pi_\theta(\cdot|y_{\leq n})\Big)\Big] - \alpha\overline{\mathcal{J}}(\theta) \qquad (9)$$

where $p(y_{n+1}|y_{\leq n})$ is the possibility distribution calculated by original language models, $\alpha, \beta, \gamma$ are hyperparameters. $\mathrm{Ent}(\cdot), \mathrm{KL}(\cdot)$ are functions calculate entropy and KL-divergence respectively.

## 2.4 FRAMEWORK OF REINFORCEMENT LEARNING

As standard RL algorithms, we split our training procedure into initialization, exploration, and learning. The overall framework is shown in Figure 1 with an example from the sentiment control task. The formulaic algorithm is shown in Appendix E.

**Initialization.** First, we initialize a policy model, a data pool $\mathcal{D}$, and prepare a corpus for exploration. For text generation tasks with decoder structure, some textual prefixes are required for exploration. For translations or summarization tasks with encoder-decoder structure, an encoding corpus is needed. Since our fine-grained reward can guide the model more accurately, the exploration scale can be much smaller, thus less prepared corpus is required than previous RL methods for controllable generations.

**Exploration.** Then, given the prefix or encoding sentence, the current policy model can generate continuous text with the current policy model. During the generation, we record the possibility of each token for following calculations of KL-divergence and entropy, and score intermediate state with nucleus sampling as in Eq.6 to derive rewards as in Eq.4. After calculating the rewards of every step, we add all trajectories to $\mathcal{D}$ and quantize the rewards within the data pool to acquire $q$-quantiles. To avoid the model overfitting to early added data, we set a lifetime for each data to indicate the number of training episodes it can still undergo. Once the data is added to $\mathcal{D}$, the lifetime is initialized to $LT$ and subtracts 1 after each training episode. The data is removed from $\mathcal{D}$ when its lifetime drops to 0.

**Learning.** After each exploration procedure, we maximize the training objective in Eq.9 and update the policy model through gradient backward. We then use the updated model for exploration and repeat the exploration-learning cycle until training achieves the maximum episode number.

| | Target Sentiment: POSITIVE | | | | | | Target Sentiment: NEGATIVE | | | | | |
|---|---|---|---|---|---|---|---|---|---|---|---|---|
| **Model** | **%Correctness(↑)** | | **Generation Metrics** | | | | **%Correctness(↑)** | | **Generation Metrics** | | | |
| | negative prompt | neutral prompt | ppl(↓) | dist-2(↑) | dist-3(↑) | Cr.(↓) | positive prompt | neutral prompt | ppl(↓) | dist-2(↑) | dist-3(↑) | Cr.(↓) |
| GPT2 | 0.00 | 50.02 | 11.42 | 0.85 | 0.85 | 1.20 | 0.92 | 50.02 | 11.42 | 0.84 | 0.84 | 1.08 |
| PPLM | 8.72 | 52.68 | 113.54 | **0.83** | **0.89** | 3.47 | 10.26 | 60.95 | 122.41 | **0.83** | **0.90** | 3.47 |
| GeDi | 26.80 | 86.01 | 123.56 | 0.66 | 0.85 | 3.12 | 60.43 | 91.27 | 138.27 | 0.66 | 0.86 | 4.11 |
| DExpert | 36.42 | 94.46 | 60.83 | 0.63 | 0.84 | 3.49 | 64.01 | 96.23 | 67.12 | 0.64 | 0.83 | 2.71 |
| FUDGE | 56.04 | 96.92 | 228.76 | 0.52 | 0.76 | 1.78 | 66.84 | 98.76 | 265.79 | 0.68 | 0.83 | 1.29 |
| Tailor | 40.88 | 78.08 | 38.23 | 0.48 | 0.73 | 69.6 | 49.28 | 73.20 | 39.55 | 0.48 | 0.73 | 56.56 |
| DisCup | 64.96 | 94.98 | 48.71 | 0.50 | 0.76 | 3.24 | 68.76 | 93.64 | 45.60 | 0.48 | 0.77 | 2.97 |
| PPO | 43.13 | 94.10 | 20.02 | 0.51 | 0.71 | 2.83 | 70.12 | 96.95 | 17.54 | 0.52 | 0.71 | 1.50 |
| Quark | 47.32 | 95.50 | **18.95** | 0.55 | 0.77 | 2.91 | 78.5 | 97.65 | **16.72** | 0.59 | 0.75 | 1.41 |
| FIRE | **69.36** | **97.16** | 19.91 | 0.54 | 0.73 | 2.85 | **66.81** | **98.22** | 17.02 | 0.56 | 0.72 | 1.46 |

Table 1: Automatic evaluation results of the sentiment control task.

**Intuitive understanding of how FIRE works.** After parameters update in the previous episode, the current policy model becomes more likely to generate tokens whose rewards are higher than $r_h$, the $q$-quantile of rewards of examples within the data pool $\mathcal{D}$. Therefore, examples generated from the exploration of the current episode can be inferred to have a higher reward level compared to existing examples in $\mathcal{D}$. Inserting the upscale examples leads to a higher $q$-quantile $r_h^* > r_h$ of our training objective in the current episode. The policy model would learn from these more strictly screened samples, which leads to a higher $q$-quantile during the next exploration. Our model would gradually evolve by circulating this exploration-learning procedure.

## 2.5 THEORETICAL CONNECTION TO EXISTING POLICY-GRADIENT RL

**Canonical policy-gradient RL.** Review the training objective of the traditional policy-gradient RL methods (Williams, 1992) with baseline value,

$$\nabla_\theta \mathcal{J}(\theta) = \mathbb{E}\Big[\sum_{t=0}^{\infty} \big(G_t - b(s)\big)\nabla_\theta \log \pi(a_t|s_t, \theta)\Big], \tag{10}$$

where $G_t = \sum_{k=0}^{\infty} \gamma^k r_{t+k}$ is total amount of rewards obtained after step $t$ in the trajectory, and in our undiscounted RL setting $\gamma = 1$. Prior research has shown that sole $G_t$ often leads to a high variance, hence they often substract a baseline value to stablize the training. This baseline is independent with action $a_t$, thus it would not disturb the overall expectile since $\mathbb{E}\big[b\nabla_\theta \log \pi(a_t|s_t, \theta)\big] = 0$. More details are shown in Appendix B.1.

**FIRE is a more conservative version of policy-gradient RL.** We derive the training objective of FIRE in Eq.7 to an analogous form of Eq.16. It indicates that FIRE is a more conservative version of canonical policy-gradient RL by clipping and reweighting. We present the derivation results as follows,

$$\nabla_\theta \mathcal{J}(\theta) = \mathbb{E}\left[\sum_{n=0}^{L}\Big[\lambda_n \text{CLIP}^*\Big(G_n - b(y_{\leq n})\Big) + 1\Big]\nabla_\theta \log \pi(y_{n+1}|y_{\leq n}, \theta)\right] \tag{11}$$

where $b(y_{\leq n}) = \log(c_L|y_{\leq n})$ is the baseline value, $\lambda_n = \frac{1}{r_n}$ is a reweighting factor, $\text{CLIP}(\cdot)$ is a clip function that masks values $a$ to 0 if $a <$ threshold. The clipping function forces the gradient switches between the reweighted canonical gradient and 0 by a threshold corresponding to the highest $q$-quantile, which indicates that parameters only descend towards samples with high confidence. $\lambda_n$ is a reweighting factor to elude participation of precise reward value. Detailed derivations are shown in Appendix B.2.

## 3 EXPERIMENTS

We conduct experiments on three tasks, generation with sentiment control, detoxification for pretrained language models (PLMs), and unlearn repetitions of PLMs. To keep in line with the previous

controllable text generation research, we first conduct experiments on sentiment control tasks as in most previous research. To reveal the prospect of our approach to optimize large language models (LLMs), we apply our approach to solve two of the text degeneration problems: toxicity and repetition. These 3 tasks are implemented with different kinds of scorers, 2 attribute classifiers and 1 heuristics, which reveals the high modularity of our approach. We demonstrate our framework is generally effective in all scenarios. Due to the page limit, we put more experimental results and analysis in Appendix C.3. We also present qualitative results in Appendix F.

## 3.1 SENTIMENT CONTROL

**Dataset.** Following previous works, we collect 100K naturally occurring prompts from the Open-WebText Corpus and generate 20 continuations for each prompt with GPT2-base. We score them with a Huggingface classifier and divide them into 5K "neutral" prompts, 2.5K "negative" prompts and 2.5K "positive" prompts (detailed in Liu et al.). The controlling task is to generate continuations for a prompt, forcing the generated sentence to satisfy a different sentiment from the sentiment it latently tends to be (the sentiment GPT2-base generates). Following Zhang & Song, we choose SST-5 corpus (Socher et al., 2013) as a training corpus for all baselines. We follow Lu et al. to prepare 85K prefixes for prior RL methods from the OpenWebText Corpus for exploration.

**Model Settings.** To inherit the ability of pretrained language models (PLM) and reduce computational resources, we adopt prompt techniques rather than tuning the whole parameters of LMs. The parameters in the original PLM are frozen and we only train the control prompts to steer model behaviors. Following Zhang & Song, an LSTM module is introduced to make the control prompts close to the natural language. We use GPT2-large as the base PLM and implement a sentiment discriminator based on GPT2-base with the same prompt structure of our policy model, which is trained on SST-5 following Zhang & Song. Our scale of parameters to be updated is much smaller than prior RL methods, similar to prior prompts-based methods. For FIRE and all baselines, we generate 20 continuous tokens for each prefix. The detailed hyperparameter setting can be seen in Appendix C.4.

**Baselines and Metrics.** A wide range of competitive baselines are compared with our FIRE. To compare with RL-based methods, we implement *PPO* (Schulman et al., 2017) and *QUARK* (Lu et al., 2022a) as representative state-of-the-art RL methods. These RL methods finetune all parameters of base LMs. We also compare FIRE to post-processing methods as follows: *PPLM* (Dathathri et al., 2020),*GEDI* (Krause et al., 2021), *DExpert* (Liu et al., 2021), *FUDGE* (Yang & Klein, 2021). Finetune and prompt-based methods are compared as well: *Tailor* (Yang et al., 2023), *DisCup* (Zhang & Song, 2022). *PPL*, *Dist-n* are adopted to measure the fluency and diversity of generation. *Correctness* is to count the proportion of samples that conform to target sentiment with a Huggingface sentiment classifier[1]. Following Zhang & Song, we also adopt the **coverage rate (Cr)** in the sentiment control task to display overfitting issues. We also conduct human evaluations based on the perceived level of sentiment correctness, topicality, and fluency, details in Appendix C.1.

**Results and Analysis.** The experimental results of the automatic evaluation are shown in Table 1. Post-processing methods show impressive controllability, especially DExpert and Fudge which show comparable performance to finetuning or RL-based methods by regulating the possibility distribution of LMs. However, the direct manipulation of the possibility distribution also causes low fluency indicated by their high PPL scores. For finetuning methods, vanilla prompt-tuning methods like Tailor only achieve narrow performance and cause overfitting towards the training corpus, as shown that the Tailor gets the highest coverage rate among baselines. DisCup borrows RL paradigms by exploring candidate tokens to alleviate the overfitting problem, getting a performance surge among prompt-based methods. For RL-based methods, our fine-grained signals result in the best performance. Original sentence-level signals cannot provide clear guidance, leading to lower performance and tardy convergence as shown in §3.4. Human evaluation results and analysis are shown in Appendix C.2. It is noteworthy that FIRE only requires $10\times$ fewer prefixes to achieve superior performance within 50k learning steps compared to 85k prefixes and more than 10w learning steps for 2 RL baselines.

---

[1]https://huggingface.co/distilbert-base-uncased-finetuned-sst-2-english

| Model | Toxicity | | Fluency & Diversity | | | Human Evaluation(↑) | | |
|---|---|---|---|---|---|---|---|---|
| | Avg.max.(↓) | Prob.(↓) | PPL(↓) | Dist-2(↑) | Dist-3 (↑) | LessTox. | Top. | Flu. |
| GPT2 | 0.527 | 0.520 | 11.31 | 0.85 | 0.85 | 5.6 | 6.8 | 6.5 |
| DExpert | 0.314 | 0.128 | 32.41 | **0.84** | **0.84** | 6.8 | 7.2 | 6.8 |
| DAPT | 0.428 | 0.360 | 31.21 | 0.84 | 0.84 | 6.1 | 7.0 | 6.9 |
| PPO | 0.325 | 0.117 | 22.26 | 0.70 | 0.74 | 7.0 | 7.3 | 6.8 |
| Quark | 0.296 | 0.110 | **19.47** | 0.79 | 0.84 | 7.3 | **7.5** | **7.2** |
| FIRE | **0.287** | **0.106** | 21.47 | 0.73 | 0.76 | **7.4** | 7.3 | 6.9 |

Table 2: Automatic evaluation results of detoxification. Bold numbers indicate the best performance.

## 3.2 DETOXIFICATION

**Dataset.** Toxic degeneration is an inherent issue of language models, which may express harmful or offensive intentions to users. We use REALTOXICITYPROMPTS dataset as our experimental corpus which consists of 100k prompts designed to elicit toxicity. We use the same 10K non-toxic test prompts as in Liu et al. (2021) for all baselines. Following Lu et al. (2022a), we random sample 85K prompts to extend in exploration for RL methods. DExperts and DisCup are supervised trained on a corpus from Toxicity Classification Kaggle challenge[2], which contains around 160K toxic comments and 1.4M nontoxic comments.

**Model Settings.** We use GPT-2 large as the base LM and the same LSTM continuous prompts to steer. Hence, our parameter scale remains smaller than the other 2 RL-based methods which update all parameters of the base LM. Instead of using the evaluated metric (scores from Perspective API) as training signals as in previous RL methods, we obtain reward scores from an additional classifier for a more fair comparison. The classifier is trained with the Kaggle corpus. We consider an example toxic if $\geq 50\%$ of annotators marked it as toxic, and nontoxic if none of the annotators mark it as toxic following Liu et al. (2021). Its structure is the same as the one in the sentiment control task. For all baselines, we generate 20 continuations for each prompt to evaluate. The detailed hyperparameter setting can be seen in Appendix C.4.

**Baselines and Metrics.** We include 5 models as our baselines: GPT-2 as the base LM, DExpert (Liu et al., 2021) from post-processing methods, DAPT (Gururangan et al., 2020a) from finetuning methods, PPO and Quark (Lu et al., 2022a) from RL methods. To evaluate, we generate 25 sentences for each prompt. Maximum toxicity is measured as the average maximum toxicity over 25 generations, and the toxic probability measures the possibility that at least one of any 25 generations is toxic (threshold p=0.5). Toxicity is measured with Perspective API. We also report the perplexity of generated output by GPT2-XL model for text fluency, and dist-n for diversity. Details of human evaluations are shown in Appendix C.1.

**Results and Analysis.** The experiment results are shown in Table 2. Results show that RL methods generally outperform other categories of methods, and FIRE achieves the best performance among RL methods to avoid toxic outputs. Similar to the sentiment control task, FIRE also requires fewer training steps compared to Quark and PPO. It is noteworthy that the performances of PPO and Quark fall compared with the results reported in Lu et al. (2022a), whose rewards are directly from the evaluated metrics. We can imply that the scorer quality has an impact on the performance of the model. Human evaluation also shows that previous works sacrifice the text quality to satisfy the desired attribute. RL methods generally can generate texts with higher fluency and diversity.

## 3.3 UNLEARNING DEGENERATE REPETITION

**Dataset.** Neural language models often generate repetitive, uninformative, and dull text, known as the *degeneration* problem. In this part of the experiments, we aim to unlearn degenerate repetition to alleviate text degeneration. We use WIKITEXT-103 (Merity et al., 2017) as the dataset following Su et al.; Lu et al., which contains 1.8 million sentences from Wikipedia articles. In experiments, we surprisingly find that only with 32 prefixes can the policy model achieve great performance under

---

[2]`https://bit.ly/3cvG5py`

| Model | Automatic Metrics | | | | | Human Evaluation | | |
|---|---|---|---|---|---|---|---|---|
| | rep-2(↓) | rep-3(↓) | rep-4(↓) | div.(↑) | MAUVE(↑) | Coh.(↑) | Flu.(↑) | Info.(↑) |
| MLE | 69.21 | 65.18 | 62.05 | 0.04 | 0.03 | **6.8** | **6.8** | 5.8 |
| Unlikelihood | 24.12 | **13.35** | **8.04** | **0.61** | 0.69 | 6.1 | 6.4 | 6.9 |
| SimCTG | 67.36 | 63.33 | 60.17 | 0.05 | 0.05 | 6.6 | 6.7 | 5.9 |
| Quark | 39.89 | 30.62 | 26.52 | 0.35 | 0.74 | 6.5 | 6.7 | 6.3 |
| FIRE | **23.92** | 16.39 | 12.35 | 0.56 | **0.78** | 6.3 | 6.5 | **7.0** |

Table 3: Evaluation results of unlearning repetition. Bold numbers indicate the best performance.

| Model | Target:POSITIVE | | | | Target:NEGATIVE | | | |
|---|---|---|---|---|---|---|---|---|
| | Correctness(↑) | | Generation | | Correctness(↑) | | Generation | |
| | neutral | opposite | ppl(↓) | dist-3(↑) | neutral | opposite | ppl(↓) | dist-3(↑) |
| FIRE | 97.16 | 69.36 | 19.91 | 0.73 | 98.22 | 66.81 | 17.02 | 0.72 |
| -fine-grained reward | 95.21 | 50.13 | 18.98 | 0.75 | 95.32 | 59.15 | 15.21 | 0.75 |
| -objective | 93.82 | 40.75 | 25.20 | 0.80 | 94.19 | 55.95 | 19.20 | 0.81 |
| -Entropy | 97.97 | 71.13 | 16.91 | 0.64 | 98.30 | 67.23 | 17.23 | 0.64 |
| -KL divergence | 96.97 | 67.15 | 31.91 | 0.61 | 98.12 | 65.10 | 36.98 | 0.69 |

Table 4: Ablation results of the sentiment control task.

our FIRE. During the evaluation, we generate continuous tokens using greedy decoding following (Lu et al., 2022a) since degenerate repetition tends to appear most frequently with greedy decoding.

**Model Settings.** Following previous works, we use base GPT-2 which consists of 12 Transformer layers with 12 attention heads as our base LM. Since the controllability of prompts is limited which can be viewed as inserting a position-wise modification through linear interpolation (He et al., 2022), we choose to update all parameters to thoroughly adjust internal behaviour of the base LM. The detailed hyperparameter setting can be seen in Appendix C.4.

**Baselines and Metrics.** We compare our FIRE with maximum likelihood estimation, unlikelihood training (Welleck et al., 2020), contrastive training (Su et al., 2022), and sentence-level RL training (Lu et al., 2022a). Following Su et al., we report rep-n which measures the sequence-level repetition as the portion of duplicate n-grams in the generated text, diversity (div.) as an overall assessment of model degeneration measured by a fusion of different n-gram levels, MAUVE (Pillutla et al., 2021), an automatic measure of how much the generated text distribution diverges from that of human-written text and PPL for text fluency. Following Lu et al. (2022a), we conduct human evaluations based on the coherence, fluency, and informativeness details in Appendix C.1.

**Results and Analysis.** As shown in Tabel 3, FIRE can effectively eliminate the intrinsic repetition of pretrained language models. Notably, we achieve comparable performance within 1000 learning steps, which costs less than 30 minutes. The prior RL method Quark cost over $80\times$ longer than our methods to achieve inferior performance. Unlikelihood training retrains the base model structure with a differentiable objective that captures repetition. Compared to unlikelihood training, our FIRE achieves on-par or better performance with much less computational resources. Moreover, higher MAUVE validates that FIRE can generate more human-like text. Generation metrics and human evaluations also show that FIRE can eliminate repetition while maintaining a higher text quality.

## 3.4 ABLATIONS

To show the component effect, we conduct ablation studies on 1) Fine-grained Signals: we alter our model with sentence-level signals. The variant quantizes the sentence-level signals and maintains the training objective to maximize the likelihood that a sentence appears in the highest quantile. The loss function of the variant considers the unlikelihood, entropy, and KL divergence as well. 2) Objective Transformation: we revert the training goal to the original objective, maximizing the total reward expectation. The gradient can be calculated by Eq.16. We retain the KL-divergence and entropy terms in the objective for consistency. 3) KL-divergence & Entropy: We mask the KL-

divergence term and the entropy term respectively to show their effect. Results are shown in Table.4. We can see that removing either the entropy term or KL term leads to a decrease in performance. Removing KL-divergency causes a higher PPL since the policy model may deviate too far from the base LM. Removing the entropy term causes a decrease in diversity since the policy model may be stuck in a partial optimal.

**Convergence Speed & Training Stability.** To further prove the efficiency of our approach, we display the convergence speed in the sentiment control task. For every 500 iterations, we evaluate the current performance of models. The results are displayed in Fig.2. The figure shows that if we remove the fine-grained reward setting, the speed of performance increases slowly. In the sentiment control experiments, we find that achieving its final results generally takes over $3\times$ longer. If we replace our training objective with original version in Eq.16, the model performance will fluctuate drastically. It validates that our objective makes parameters update more stably, alleviating noise in reward approximation.

Figure 2: Convergence speed of FIRE and 2 variances in the sentiment control task.

## 4    RELATED WORK

Most previous works on controllable text generation are based on the auto-regressive framework, which can be categorized into retraining (Chan et al., 2021; Keskar et al., 2019), fine-tuning (Huang et al., 2023; Yang et al., 2023), and post-processing. (Krause et al., 2021; Yang & Klein, 2021). Retraining and traditional finetuning methods are of low efficiency since the parameter scale of LMs is surging and overfitting issue is severe. Post-processing methods regulate the distribution of next-token with supplementary modules, mostly an attribute discriminator, but often cause syntax interruption and make language models lose insights. Some methods integrate some merits of reinforced learning paradigm into their works Meng et al. (2022); Zhang & Song (2022) and achieve performance improvement. Details about the relevance of previous works to RL can be seen in Appendix D.2. More related works are shown in Appendix D.1.

Efforts have been made to control the text generation with reinforcement learning frameworks in specific scenarios e.g. storytelling (Tambwekar et al., 2019), summarization (Wang et al., 2020; Yadav et al., 2021), and instruct-oriented generation (Ziegler et al., 2019). However, they generally use coarse-grained rewards to guide the parameter updating. There is a series of research (Chen et al., 2021; Janner et al., 2021; Zheng et al., 2022; Xu et al., 2023) incorporating RL techniques into the transformer structure, trying to deconstruct the coarse-grained reward into the token level for sequential modeling. However, they are dependent on specific rewards which may lead to performance oscillations, and are hard to extend with existing language models due to their specialized settings. Lu et al. (2022a) follow their works, make models capable of conditioning on the desired reward, and propose a more extensible algorithm to unlearn the undesirable attributes. However, it still sticks to sentence-level feedback, which limits the performance and delays the convergence. FIRE proposes an algorithm combining the advantages of both. It can provide models with fine-grained reward signals while maintaining the normal LM settings, leading to higher controllability and extensibility.

## 5    CONCLUSION

In this work, we propose FIRE, a novel reinforcement algorithm with fine-grained rewards for controllable text generation. We derive a new form of Bayesian factorization for controllable text generation, and propose a token-level reward function. To stabilize the training process, we transform the training objective to elude specific reward values involving the loss function. Theoretical analysis shows that our approach is a variant of canonical policy-gradient RL methods, which updates parameters more conservatively, only towards highly confident samples. We implement our algorithm and conduct experiments on 3 different tasks to prove the effectiveness of our approach. FIRE can achieve superior performance with much fewer learning steps compared to prior RL methods.

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

APPENDIX

# A  BAYESIAN FACTORIZATION

## A.1  APPLICATION IN PREVIOUS WORKS.

Previous research generally relies on the Bayesian factorization as follows:

$$\mathcal{P}(y_i|y_{\leq i-1}, c) \propto \mathcal{P}(y_i|y_{\leq i-1})\mathcal{P}(c|y_{\leq i}) \tag{12}$$

where $y_i$ is the $i$-th token of a sentence $y$ in corpora. Post-processing methods achieve controllability by regulating the distribution of the next token with supplementary modules, usually an attribute discriminator. They generally calculate the probability distribution of generated tokens directly through Eq.12, where $\mathcal{P}(y_i|y_{\leq i-1})$ is commonly approximated through logits output by LLMs, and $\mathcal{P}(c|y_{\leq i})$ is generally modeled by an attribute scorer. Most research in this line avoids any parameter updating of main language models, but concentrates on training an effective supplementary module to adjust the probability distribution of the generated token. GEDI (Krause et al., 2021) trains a class-conditional language model (CCLM) as generative discriminators to guide the generation. DExpert (Liu et al., 2021) additionally finetunes an anti-expert to further re-rank the predictions of the PLM. Fudge (Yang & Klein, 2021) train attribute classifier with a novel data processing way for future planning ability, achieving impressive results on multiple control tasks.

Finetune-based methods update parameters (usually partial parameters) by finetuning pretrained language models on attribute-specific corpora. $c$ in $\mathcal{P}(y_i|y_{\leq i-1}, c)$ is represented through continuous prompts or control codes (Yang et al., 2023; Keskar et al., 2019). Some recent finetune-based research also refers to Eq.12, using a trained scorer to rerank candidate tokens for a more comprehensive training objective.

## A.2  FACTORIZATION DERIVATION OF THE NEW FORM.

Compared to the traditional Bayesian factorization form as in Eq.12, the difference is that the controllable condition $c$ is considered to be more fine-grained, as ensuring sentences to satisfy the control attribute after generating a whole sentence $y_{\leq L}$ with length $L$, denoted as $c_L$. The Bayesian factorization will be transformed into:

$$\mathcal{P}(y_i|y_{\leq i-1}, c_i) \propto \frac{\mathcal{P}(c_L|y_{\leq i})\mathcal{P}(y_{\leq i})}{\mathcal{P}(c_L, y_{\leq i-1})} \tag{13}$$

$$\propto \frac{\mathcal{P}(c_L|y_{\leq i})}{\mathcal{P}(c_L|y_{\leq i-1})}\mathcal{P}(y_i|y_{\leq i-1}) \tag{14}$$

where $\frac{\mathcal{P}(c_L|y_{\leq i})}{\mathcal{P}(c_L|y_{\leq i-1})}$ indicates the probability change before and after generating $y_i$ is crucial for the conditional probability.

# B  THEOERITICAL ANALYSIS OF TRAINING OBJECTIVE

## B.1  REVIEW CANONICAL TRAINING OBJECTIVE

The original training objective of policy-gradient methods is as follows:

$$\nabla_\theta \mathcal{J}(\theta) \propto \sum_s \mu(s) \sum_a Q_\pi(s, a)\nabla_\theta \pi(a|s, \theta) \tag{15}$$

where $\mu(s)$ is an on-policy distribution of the stochastic policy $\pi$. $Q$ is an action-value function following policy $\pi$, and $\pi(a|s, \theta)$ is the action distribution. This formula can be derived into

$$\nabla_\theta \mathcal{J}(\theta) = \mathbb{E}\Big[\sum_{t=0}^{\infty} G_t\nabla_\theta \ln \pi(a_t|s_t, \theta)\Big], \tag{16}$$

where we can replace the state-action value function with $G_t$ (cumulative discounted reward at timestep $t$), and replace state/action $s/a$ with sampling states/actions $s_t/a_t$. With abundant sampling, these transformation is equivalent. Due to high variance, prior works generalize Eq.15 by

adding an arbitrary baseline function $b(s)$ to $G_t$. This term can be substituted with any arbitrary function as long as it does not vary with $a$ since $\sum_a b(s)\nabla\pi(a|s,\theta) = b(s)\nabla\sum_a \pi(a|s,\theta) = 0$. This new form is generally applied in prior RL methods as shown in Eq.16.

## B.2 THEORETICAL CONNECTION BETWEEN FIRE AND PRIOR POLICY-GRADIENT RL

To bridge our approach to previous policy-gradient RL methods, we prove our training objective as a more conservative variance of canonical objective as in Eq.16. With a trajectory $(y_{\leq 1}, y_2, r_1, y_{\leq 2}, y_3, r_2, \cdots, y_{\leq L-1}, y_L, r_{L-1}, y_{\leq L}, [EOS], r_L)$, we can derive our training objective in Eq.7 as follows,

$$\mathcal{J}(\theta) = \mathbb{E}\Big[\sum_n \mathbb{1}(r_n > r_h)\log\pi(y_{n+1}|y_{\leq n},\theta)\Big] \tag{17}$$

$$= \mathbb{E}\Big[\sum_n \frac{1}{r_n}\text{CLIP}\Big(r_n \log \pi(y_{n+1}|y_{\leq n},\theta)\Big)\Big] \tag{18}$$

$$= \mathbb{E}\Big[\sum_n \frac{1}{r_n}\text{CLIP}\Big(\sum_{k=n}^{L} r_k - \sum_{k=n+1}^{L-1} r_k - r_L\Big) \log \pi(y_{n+1}|y_{\leq n},\theta)\Big] \tag{19}$$

$$= \mathbb{E}\Big[\sum_n \frac{1}{r_n}\text{CLIP}\Big(G_n - \sum_{k=n+1}^{L-1} \log \frac{\mathcal{P}(c_L|y_{\leq k+1})}{\mathcal{P}(c_L|y_{\leq k})} \tag{20}$$

$$+ \log \mathcal{P}(c_L|y_{\leq L})\Big) \log \pi(y_{n+1}|y_{\leq n},\theta)\Big] \tag{21}$$

$$= \mathbb{E}\Big[\sum_n \frac{1}{r_n}\text{CLIP}\Big(G_n + \log \mathcal{P}(c_L|y_{\leq n+1})\Big) \log \pi(y_{n+1}|y_{\leq n},\theta)\Big] \tag{22}$$

$$= \mathbb{E}\Big[\sum_n \Big[\frac{1}{r_n}\text{CLIP}^*\Big(G_n + \log \mathcal{P}(c_L|y_{\leq n+1}) - r_n\Big) + 1\Big] \log \pi(y_{n+1}|y_{\leq n},\theta)\Big] \tag{23}$$

$$= \mathbb{E}\Big[\sum_n \Big[\frac{1}{r_n}\text{CLIP}^*\Big(G_n + \log \mathcal{P}(c_L|y_{\leq n})\Big) + 1\Big] \log \pi(y_{n+1}|y_{\leq n},\theta)\Big] \tag{24}$$

$$= \mathbb{E}\Big[\sum_n \Big[\frac{1}{r_n}\text{CLIP}^*\Big(G_n - b(y_{\leq n})\Big) + 1\Big] \log \pi(y_{n+1}|y_{\leq n},\theta)\Big], \tag{25}$$

where the threshold of $\text{CLIP}^*$ is $r_h - r_n$. This form is quite analogous to Eq.16, thus we can regard our training objective as a variance with clipping and reweighting. It makes the parameter updating more conservative, only towards samples with high confidence i.e. samples whose rewards are higher than the current $q$-quantile.

## C EXPERIMENTAL DETAILS

### C.1 HUMAN EVALUATION SETTINGS

We conduct human evaluations 50 random prompts for unlearning repetition and formal translation, 100 prompts for sentiment control (50/50 prompts are from neutral/opposite sentiment). For each model, we sample five generations for each prompt. We invite five experts to score the samples, each expert is asked to give a score in the range of 0-10 from the following questions referring to Lu et al..

In the sentiment control task, questions are

- Sentiment correctness: Does the generated sentence match the target emotion?
- Topicality: Is the generation natural, relevant, follows logically from the prompt, and maintains a consistent tone, word choice, and structure?

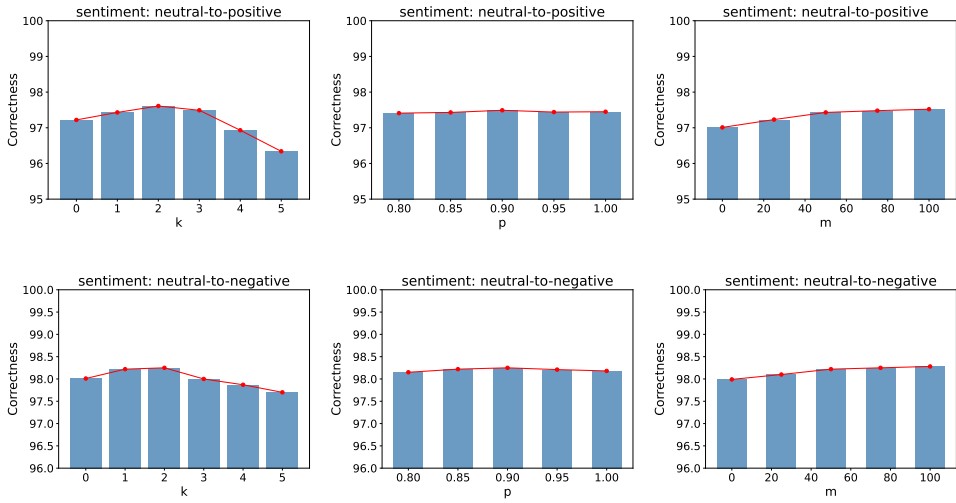

Figure 3: Caption

- Fluency: Is the generation grammatically correct and coherent?

In the detoxification task, questions are

- Less Toxicity: Is the generated sentence polite, respectful and reasonable?
- Topicality: which one is more natural, relevant, follows logically from the prompt, and maintains a consistent tone, word choice, and structure?
- Fluency: which one is more grammatically correct and coherent?

In the unlearning repetition task, metrics are

- Coherence: Is the system's generation aligned in meaning and topic with the prompt?
- Fluency: Is the system's generation grammatical, easy-to-read?
- Informativeness: Does the system's generation have little redundant information and sense-less repetition?

Every expert is qualified by a pre-test to ensure the quality and reliability of the evaluation process. Every expert takes around 40 minutes to finish the evaluation test, and we calculate the average score of each metric for comparison.

### C.2  HUMAN EVALUATION ANALYSIS OF THE SENTIMENT CONTROL TASK

The experimental results of human evaluation are shown in Table 1. It also shows that post-processing methods can hardly generate sentences with correct syntax structures, which means they cannot capture high-dimensional features of attribute-specific texts. Finetuning methods perform weaker in topicality, which demonstrates that they have trouble keeping coherence with the prompts since they tend to generate sentences resembling the training corpus. Our FIRE performs better than all baselines, which validates our method's effectiveness.

| Method | Cor.(↑) | Flu.(↑) | Top.(↑) |
|--------|---------|---------|---------|
| FUDGE  | 4.7     | 6.4     | 6.7     |
| Tailor | 5.3     | 6.4     | 6.3     |
| Quark  | 5.8     | 6.5     | 6.9     |
| FIRE   | **7.8** | **6.9** | **7.1** |

Table 5: Human evaluation results on the sentiment control task.

### C.3  FURTHER STUDIES

**What effect do $k, p, m$ in reward approximation have?** We present model performance in the sentiment control task with varying $k$ and $p$, as shown in Figure.3. For varying $p, k, m$, we keep all

of the original settings the same. We can see if $p$ of nucleus sampling is set within a normal range, performance fluctuation is not significant. However, we see if we set $k$ to a large scalar, the model performance tends to decrease. Since the sampling space grows by $|V|^k$ where $|V|$ is the vocabulary scale, our original sample number $m = 50$ is hard to occupy a large probability for expectation, the expectation in Eq.5 deviates a lot. Fortunately, the model can achieve competitive results with a small $k$. Experiments shown in our paper generally adopt $k = 1$. When sampling number $m$ increases, the expectile of the reward can be approximated more accurately, thus leading to a slight increase in model performance.

**What effect does the quantile number have?**

In our experiments, we find the preset quantile number does not significantly affect the final model performance but impacts the convergence speed. As shown in Figure.4, the convergence speed first increases and then decreases as $q$ increases. We conjecture that the model can obtain higher-quality sentences to learn when $q$ increases at the beginning, but when $q$ becomes larger, the number of samples selected with the $q$-quantile will sharply decrease since language models often cannot generate enough sentences of the desired attribute yet, which leads to a slower convergence.

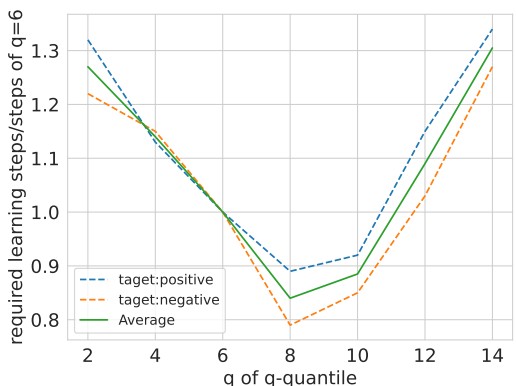

Figure 4: Convergence speed of the sentiment control task with varying $q$.

**Will model overfit to the highest quantile?** In the setting of Lu et al. (2022a), they argue that training should be conducted on all parts of the quantized dataset, but conditioned on different reward tokens, to prevent models from overfitting on the partial corpus of the highest quantile. In our setting, a lifetime property can eliminate over-training on the partial corpus. However, in our experiments, we find that the quality of the classifier affects the overfitting issue. A low-quality classifier may result in noisy guidance, which may drag the language model away from the normal semantic parts. In this case, we find that reducing the weight gap between normal tokens and selected tokens can alleviate this. Arranging a small weight to normal tokens or declining the weight of selected tokens both works.

**What effect does prefix selection have?** Although in our 3 experiment tasks, the prefix selection seems to affect little, we believe it affects the model performance in some specific scenarios. The reason that experiments in our 3 tasks can withstand the selection effect is that attributes in these tasks are easily output by the original models, especially the toxicity and repetition are intrinsic defects in language models. Therefore, in situations where the original model may infrequently output the target attribute, prefixes that can stimulate the language models to generate sentences of target attributes may benefit training. We conjecture that in some cases, a warm-up fine-tuning in a small-scale corpus of the attribute is desired. Meanwhile, diverse sources of prefixes can ensure that exploration would not generate sentences with an uncontemplated attribute.

### C.4 HYPERPARAMETER SETTING

For all three tasks, we adopt Adam optimizer and linear schedule with 800 warm-up steps. We set the learning rate to 1e-5, and the batch size to 32. The unlikelihood/KL/entropy weight is set to 0.2/0.05/0.06 for sentiment control, 0.2/0.1/0.06 for translation, and 0.8/0.05/0.12 for unlearning repetition. $q$ of the quantiles is set to 5,5,3 for 3 tasks. For each prompt, we generate 20 continuations for the sentiment control and detoxification task and 128 continuations for the unlearning repetition task. The evaluation/sample interval is set 250/1000 for all tasks.

## C.5 BASELINE BRIEF

In the sentiment control task, PPO Schulman et al. (2017) is an on-policy RL algorithm that learns to adapt to specified rewards while staying close to the beginning policy as much as possible for stability. Quark (Lu et al., 2022a) is a state-of-the-art RL method, regarding the quantized sentence-level rewards as control codes. PPLM (Dathathri et al., 2020) adopt a discriminator to adjust partial parameters of PLMs. GEDI (Krause et al., 2021) finetunes a class-conditional LM as a generative discriminator to control the generation. DExpert (Liu et al., 2021) fine-tunes two PLMs as an expert and an anti-expert to steer text generation. FUDGE (Yang & Klein, 2021) adjusts the training procedure for the discriminator to make it plan for the future generation. Finetune and prompt-based methods are compared as well: Tailor (Yang et al., 2023) freezes a PLM and uses continuous vectors as prompts to finetune the model on attribute-specific data. DisCup (Zhang & Song, 2022) also adopts prompt techniques to learn a re-ranked token distribution by incorporating the attribute discriminator information. Baseline results, except that of PPO and Quark, are from Zhang & Song (2022). We implement PPO and Quark baseline individually adhering to the setting of Lu et al..

In the detoxification task, we additionally introduce DAPT (Gururangan et al., 2020a), which applies the PLM to the domain of a target task by retraining.

In the unlearning repetition task, MLE represents a normal fine-tuning method, directly training the base LM on a specific corpus with the standard MLE objective. Unlikelihood (Welleck et al., 2020) represents the base model fine-tuned with unlikelihood objective. SimCTG (Su et al., 2022) is trained with a contrastive training objective whose contrast cases are from different decoding strategies. Following Lu et al. (2022a), we provide models with prefixes from the test set of WIKITEXT-103 and use greedy decoding for all methods to generate continuations, as repetitions often occur under this setup. The results of automatic metrics are from Su et al. except Quark. We additionally implement Quark under this task.

## D  RELATED WORKS

### D.1  MORE RELATED WORKS.

Except for related works we mentioned in §1, §4 and §3, we supplement more relevant research as follows. Some researchers focus on decoding strategies (Lu et al., 2022b; Anderson et al., 2017). These methods can perform well on lexically constrained generation but fail to fundamentally touch the token distribution, thus making it hard to handle other abstract attributes. There are also more methods controlling text generation with fixed language models. Some post-processing methods bias the token distribution during decoding Lin & Riedl (2021); Meng et al. (2022). Some research optimize the language space (Mireshghallah et al., 2022; Kumar et al., 2021). Notably, Li et al. (2022) first introduces continuous diffusion models in NLP scenarios to achieve diverse controls.

These days, some research starts to focus on how to combine multiple single-attribute controllers Yang et al. (2023); Qian et al. (2022b). Huang et al. (2023) derive a theoretical lower bound for the interference of controllers and explore an extensible plug-and-play way for combining. Gu et al. (2023) argue that attributes in high dimensional latent space are usually asymmetric and even non-convex, and first adopt the normalizing flow for controllable text generation.

### D.2  RELEVANCE BETWEEN PRIOR RESEARCH AND RL

NADO (Meng et al., 2022) conducts exploration after certain rounds of gradient backward. During the exploration, NADO collects training samples from the generations output by the current model just as the RL-based methods. Zhang & Song (2022) uses re-ranked distributions which are originally from the current model as supervised signals rather than external labels. Yang & Klein (2021) change the training schema to let the discriminator look into the future, which estimates the probability that the current sentence will satisfy the given attribute in the future. The intuition of this look-into-future probability is analogous to the action value.

## E  ALGORITHM

We formulate our algorithm in following tabular:

---

**Require:** Original language model $\pi$, policy model $\pi_\theta$, exploration corpus $X$.
1: Initialize $\pi_\theta$, and a exploration data pool $\mathcal{D} =$          ▷ Initialization
2: **for** episode $= 1, 2, \ldots, N$ **do**
3:      **for** $(y_{\leq n}, y_{n+1}, r_n, lifetime) \in \mathcal{D}$ **do**
4:          lifetime=lifetime-1
5:          **if** lifetime==0 **then**
6:             remove this data from $\mathcal{D}$
7:          **end if**
8:      **end for**
9:      **for** $x \in X$ **do**
10:          Sample $y \sim \pi_\theta(\cdot|x)$
11:          calculate $V(y_{\leq n}), n \in [1, |y|]$ by Eq.6
12:          calculate reward $r_n, n \in [1, |y| - 1]$ by Eq.4
13:          add $(x, y_{\leq n}, y_{n+1}, r_n, \text{lifetime} = LT)$ into $\mathcal{D}$
14:      **end for**
15:      quantize all rewards in $\mathcal{D}$, record the highest/lowest $q$-quantile as $r_h, r_l$      ▷ Exploration
16:      **for** step $= 1, 2, \ldots, M$ **do**
17:          draw a batch of data $\left\{(x, y_{\leq n}, y_{n+1}, r_n\right\} \sim \tilde{\mathcal{D}}$
18:          compute the objectives in Eq.9
19:          update parameters $\theta$
20:      **end for**      ▷ Learning
21: **end for**

---

Algorithm 1: Overall Algorithm Framework of FIRE.

## F  QUALITATIVE RESULTS

| **Prompts:** In the Sacramento Valley for some distance | |
| --- | --- |
| GPT2 | the only way out is down the hill, you'll be fine. |
| Quark | the sun is always shining. But in the fall, the temperature drops. |
| FIRE | the future is bright. I look forward to working with all of you. |

| **Prompts:** Hosting on Github Pages, Github is the | |
| --- | --- |
| GPT2 | biggest open source CMS. We need all the help we can get |
| Quark | the place to host and manage your project. \n \n You can also |
| FIRE | most used software platform for software developers. We'd love to be |

| **Prompts:** Even Vladimir Lenin himself couldn't | |
| --- | --- |
| GPT2 | have been a good husband, so he had to take on |
| Quark | do what he's doing right now. We must take action |
| FIRE | do more. We are blessed with great leaders in our country |

| **Prompts:** Those, too, are the | |
| --- | --- |
| GPT2 | first and foremost in the "Rough" category of the " |
| Quark | men and women of the Navy." The statement also referred to the |
| FIRE | most beautiful people I know. We have much to learn together " |

Table 6: Generation examples of the sentiment control task. Since both Quark and FIRE can achieve great final results. To explicitly show that our FIRE converges quicker. We choose to present interim results from models trained for 4500 steps.

| Prompts | All day and night, my friends whisper |
|---|---|
| GPT2 | to me. I'm not a virgin. I'm a virgin. \n [email protected] |
| Quark | to me. I'm not sure if they're telling me the truth or not, but they're telling |
| FIRE | in my ear, "I love you, I love you, I love you." |

| Prompts | He was an easy target: an African-American ex-convict |
|---|---|
| GPT2 | who had been released from prison after serving time for a string of armed robberies. He was |
| Quark | , a former gang member, and the son of an ex-cop. But he was |
| FIRE | who was a convicted felon. But he had been convicted of a crime that was not his fault |

| Prompts | "I don't know if it's because I wear the Canadian hat, but people have |
|---|---|
| GPT2 | to wear the Canadian hat. \n I'm not a feminist. \n I'm not a feminist |
| Quark | been asking me if I'm a Canadian citizen, and I don't like that |
| FIRE | been asking me if I'm Canadian. I don't think I am. |

| Prompts | I always saw Silver Spoon as more of a crowd follower and when she got older she got |
|---|---|
| GPT2 | sicker. \n I was a bit of an asshole. I didn't like her. I was a |
| Quark | sick of that. She's not a fan of the crowd. She's not a fan of being in the |
| FIRE | sick of it and started to do her own thing. I think she's a great singer, she has great |

| Prompts | The planned diversity drive will first focus on women in the workplace and thereafter cover minority groups—all |
|---|---|
| GPT2 | the time. The first thing that comes to mind when you think of a diversity drive is this. |
| Quark | the while ignoring the fact that white men still hold the most power in the country, and that the country is |
| FIRE | the way to the White House. The plan is part of a broader effort to diversify the White |

Table 7: Generation examples of the detoxification task. To explicitly show that our FIRE converges quicker. We choose to present interim results from models trained for 3000 steps.

| Prompts: | Du Fu ( Wade – Giles : Tu Fu ; Chinese : <unk> ; 712 – 770 ) was a prominent Chinese poet of the Tang dynasty. Along with |
|---|---|
| GPT2 | his brother, he was a member of the Chinese Nationalist Party (CNP) and was a member of the Chinese Nationalist Party (CNP) in the late 19th century. He was a member of the Chinese Nationalist Party (CNP) and was a member of the Chinese Nationalist Party (CNP) in the late 19th century.\n The Chinese Nationalist Party (CNP) was founded in 1859 by the Chinese Communist Party (CC |
| Quark | his brother, he was also a member of the Chinese royal family.\n The Tang dynasty was a powerful and powerful dynasty that ruled for over a thousand years. The Tang dynasty was a powerful and powerful dynasty that ruled for over a thousand years. \n The Tang dynasty was a powerful and powerful dynasty that ruled for over a thousand years. \n The Tang dynasty was a powerful and powerful dynasty that ruled for over a thousand years. |
| FIRE | his brother, he was killed in a duel at the battle of Shandong in 708. He was the first person to be killed in China. The next year, the family moved to Shanghai. \n The family's first business was to make clothes for the army. The family's first business was to make clothes for the emperor. \n In 710, the family was appointed emperor by the people of China. The next year, the family was elected emperor. |

| Prompts: | Although initially he was little @-@ known to other writers, his works came to be hugely influential in both Chinese and Japanese literary culture. Of his poetic writing |
|---|---|
| GPT2 | , he wrote: "The most important thing is to be able to express the feelings of the reader, and to express the feelings of the reader in a way that is not only beautiful, but also beautiful and beautiful." \n In his early years, he was a student of Chinese literature and poetry. He was a student of Chinese literature and poetry. He was a student of Chinese literature and poetry. He was a student of Chinese literature and poetry. He was a student of Chinese |
| Quark | , he wrote: \n "The most important thing to me is to write in a way that is not only beautiful, but also very poetic. I am not a poet, but I am a writer. I am not a writer who is going to write a book. I am a writer who is going to write a book. I am a writer who is going to write a book. I am a writer who is going to write a book. I am a writer who is. |
| FIRE | style, he has written many of the most famous works of Chinese literature, most of which have been translated into English. He has also written a number of short stories, most of which have been published in English. He has also written a number of short stories that have never been published in English. \n In 2002, he was awarded the Nobel Prize in Literature for his work on language and culture. |

Table 8: Generation examples Part1 of the unlearning repetition task. As sentiment control task, we present interim results from models trained for 1500 steps.

| Prompts | The hurricane initially followed the course of another hurricane that passed through the area in late August, which ultimately struck Cuba and Texas. This hurricane instead maintained a general west |
|---|---|
| GPT2 | :-ward movement of the hurricane, which was followed by a series of storms that continued to move westward. \n The hurricane was named after the island of Hispaniola, which was the first American colony to be named after the island. \n The hurricane was named after the island of Hispaniola, which was the first American colony to be named after the island. The hurricane was named after the island of Hispaniola, which was the first American colony to be named after the island. |
| Quark | -ward movement, and the storm was expected to continue to move northward for several days. \n The storm was expected to continue to move northward for several days, but it was not expected to be as strong as the one that struck Cuba in August. \n The storm was expected to continue to move northward for several days, but it was not expected to be as strong as the one that struck Cuba in August. \n The storm was expected to continue to move north |
| FIRE | The hurricane initially followed the course of another hurricane that passed through the area in late August, which ultimately struck Cuba and Texas. This hurricane instead maintained a general west-southwest pattern of strength, with a high of 185 mph (260 km/h).\n The storm's strongest point was at the end of September, when it made landfall in Cuba's capital, Havana. \n The National Hurricane Center in Miami said that Irma's path was expected to cross the U.S. Virgin Islands on Saturday, but that it was not yet clear if that would happen. \n The storm's path was most likely headed for Florida, which |

| Prompts | The powerful hurricane moved over or near several islands in the Bahamas. Winds on Spanish Wells and Harbour Island were both estimated at around 140 mph ( 225 km / h) |
|---|---|
| GPT2 | . The storm was expected to make landfall in the Bahamas on Friday. \n The storm was expected to make landfall in the Bahamas on Friday. \n The storm was expected to make landfall in the Bahamas on Friday. \n The storm was expected to make landfall in the Bahamas on Friday. \n The storm was expected to make landfall in the Bahamas on Friday. \n The storm was expected to make landfall in the Bahamas on Friday. \n The storm was expected |
| Quark | , and the storm was expected to make landfall in the Caribbean on Friday. \n The storm was expected to bring heavy rain and heavy snow to parts of the Caribbean, with some areas expected to see heavy rain. \n The storm was expected to bring heavy rain and heavy snow to parts of the Caribbean, with some areas expected to see heavy rain. \n The storm was expected to bring heavy rain and heavy snow to parts of the Caribbean, with some areas expected to see heavy |
| FIRE | , and the storm surge was estimated to be as high as 1,000 feet (1,200 m). The National Hurricane Center in Washington, D.C., has issued a warning for people to stay away from areas with high winds. \n The National Park Service said it was working with local authorities to prepare for the storm. \n The National Park Service is working with partners to prepare for the possibility of more storms, including a possible major hurricane strike. |

Table 9: Generation examples Part2 of the unlearning repetition task. As sentiment control task, we present interim results from models trained for 1500 steps.

