# OpenReview forum: "Reinforcement Learning with Fine-grained Reward for Controllable Text Generation"
_ICLR.cc/2024/Conference — ICLR 2024 Conference Withdrawn Submission_

### Official Review · Reviewer_J25V · 2023-10-17

**Soundness:** 2 fair
**Presentation:** 1 poor
**Contribution:** 2 fair
**Rating:** 3
**Confidence:** 4

**Summary:**

For the safety in real-world applications, it is desirable to control the generations of the LLMs. To avoid the exposure bias between training and inference, RL methods naturally come into play.
However, the feedback in RL formulation of LM training is generally sparse, which lowers the training efficiency and stability.
To have a better control on the RL-based LM training, this paper proposes an fine-grained reward function based on Bayesian factorization.
Furthermore, to stabilize the training process, the fine-grained rewards are used in a filtering-style objective to train the LM.
On a more theoretical aspect, the authors show that the proposed method is a more conservative policy-gradient RL methods.
In the experiments over three tasks, the proposed method shows competitive performance against the baselines.

**Strengths:**

1. The overall direction of learning a dense feedback for language tasks is meaningful and promising.
2. The proposed method shows competitive performance in the experiments.

**Weaknesses:**

1. Unclear contribution of the proposed method. The authors ought to give proper discussions, citations, and credit to prior works in order to clarify the contribution. Some examples are as follows:
    - Prior works in NLP have already identified the problem of coarse-grained feedback and have proposed methods to learn a finer-grained reward function based on the coarse-grained feedback in a general setting, see, e.g., [1].
    - Since the proposed method requires a "practical classifier" $f$, it potentially assumes the availability of sufficient supervised data. With this assumption, more classical Inverse RL methods have already been applied in the prior NLP works for estimating the dense rewards, see, e.g., [2].
    - Previous NLP papers have also adopted Monte Carlo methods to rollout intermediate rewards, see, e.g., [3,4].
    - The proposed "the exploration-learning cycle" has its correspondence in prior NLP, offline RL, and RLHF papers [e.g., 1, 5,6,7,8].
    - The filtering-style objective Eq. 7 is similar to Eq. 2 and 3 in the CRR paper [9].

2. The training objective, Eq. 9, has five hyperparameter  aiming at potentially conflicting goals (three in the equation together with $r_h, r_l$), which seems to require extensive tuning and make the training objective less stable. Additionally, more ablation study should be done to reveal the effect of $\alpha$ and $r_h, r_l$.


3. The construction of the dense reward, i.e., the approximation if Eq. 6, contradicts the authors' discuss of the scorer's deviation that "traditional classifiers only provide $\log P(c_{L} \mid y_{\leq L})$". Or else, how would the authors justify such an approximation?

4. The filtering-style loss Eq. 7 & 8 may not be data efficient. Based on my understanding, it will throw away data points whose rewards are between $r_h$ and $r_l$, which could otherwise help the language-generation quality. This may be related to the relative inferior performance of the proposed method on metrics "PPL" and "Dist-n" in Table 1 and 2.

5. Given the instability of RL methods, the experiment results of the proposed method, such as those in Table 1-4, need to be accompanied with error bars.


6. This paper has many wrong or unclear terminology and formula. I strongly suggest the authors to proofread the paper to ensure the readability. Some examples are as follows:
    - In the second paragraph of Section 1: "possibility distributions", "... between training and inferring";
    - In the last paragraph of Section 1: "canonical policy-gradient RL methods" (*"canonical" has special meaning in statistics/optimization literature*), "high-confidence samples" (*there is no definition of "confidence" before this term*).
    - In Eq. 1, it needs to be $a_t \sim \pi(\cdot \mid s_t)$
    - In Eq. 2, the expectation over $s'$ is missing.
    - In "... which leads to a large deviation of the expectile", what is the definition of "expectile"? Are you referring to the "expectile" in econometrics? If so, why and how is it related?
    - Eq. 5 and 6 have cluttered notation of $y$ and $y_{\leq i}$. Also the authors may want to unified the notation for the first argument of $\pi$ to be either $a$ or $y$.
    - The formula for $b(y_{\leq n})$ below Eq. (11) is incorrect.
    - What do you mean by "continuous tokens" or "continuous text"?

***
[1] Yang, Shentao, et al. "Preference-grounded Token-level Guidance for Language Model Fine-tuning." arXiv preprint arXiv:2306.00398 (2023).

[2] Shi, Zhan, et al. "Toward diverse text generation with inverse reinforcement learning." arXiv preprint arXiv:1804.11258 (2018).

[3] Yu, Lantao, et al. "Seqgan: Sequence generative adversarial nets with policy gradient." Proceedings of the AAAI conference on artificial intelligence. Vol. 31. No. 1. 2017.

[4] Lin, Kevin, et al. "Adversarial ranking for language generation." Advances in neural information processing systems 30 (2017).

[5] Hishinuma, Toru, and Kei Senda. "Weighted model estimation for offline model-based reinforcement learning." Advances in neural information processing systems 34 (2021): 17789-17800.

[6] Yang, Shentao, et al. "A unified framework for alternating offline model training and policy learning." Advances in Neural Information Processing Systems 35 (2022): 17216-17232.

[7] Ziegler, Daniel M., et al. "Fine-tuning language models from human preferences." arXiv preprint arXiv:1909.08593 (2019).

[8] Stiennon, Nisan, et al. "Learning to summarize with human feedback." Advances in Neural Information Processing Systems 33 (2020): 3008-3021.

[9] Wang, Ziyu, et al. "Critic regularized regression." Advances in Neural Information Processing Systems 33 (2020): 7768-7778.

**Questions:**

1. The proposed method seems to consider only the bandit setting and the proposed training loss seems to be a filtered-MLE loss. Why do the authors want to formulate the proposed method in the RL framework and discuss action values and the Bellman equation?
2. Why avoid involving specific reward values into the training objective could stabilize the training process?
3. What do you mean by "updates parameters only towards high-confidence samples"?
4. I don't understand the entire sentence ends with "... since learning requires more samples of the desired attribute generated from the exploration." Why do you sample the attribute? What does "exploration" mean here? And why can it generate attribute?
5. By "quantize all rewards to obtain q-quantiles", are you quantizing among sentences corresponding to the same prefix? Or among all sentences that may correspond to different prefixes?
    - If quantization is done among all sentences, how do you ensure that the  per-step-reward distributions corresponding to different prefixes are comparable?
6. By the proposed "the exploration-learning cycle", the proposed method is basically a online RL method. Why do you think that conservatism (in Section 2.5) is beneficial?

---

### Official Review · Reviewer_bVBE · 2023-10-19

**Soundness:** 2 fair
**Presentation:** 3 good
**Contribution:** 2 fair
**Rating:** 5
**Confidence:** 4

**Summary:**

This paper presents a reinforcement learning (RL) based approach for controllable text generation for LLMs. Specifically, the authors propose use of a FIne-grained REward (FIRE) function that extends beyond usual sentence-level rewards. The authors establish a connection between their FIRE framework and traditional policy gradient based approaches, signifying that theirs conducts more conservative updates. The approach is then evaluated in 3 domains (sentiment control, detoxification, and reduction of repetitions), showing both a reduction in sample complexity and increase in final performance.

**Strengths:**

1) The presented method is simple and practical in principle, with the main trick being the approximation of the fine-grained reward by sampling in (6).

2) Experiment results are fairly solid as well, showing that the method is able to beat out vanilla RLL (e.g., PPO) approaches and several other baselines across most of the datasets.

3) The paper is well written and the contribution / technical details are fairly clear.

**Weaknesses:**

1) Arguably, the biggest weakness of fine-grained reward approaches is their computational complexity, which is traded off with sample complexity. As stated by the authors, the computational expense of computing (5) is high given that most scorers are trained at sentence-level (and not token level). The authors propose a workaround for this using (6), but the practical computational expense of this approximation remains ambiguous. It's clear in Figure 2 that the sample complexity of FIRE is preferable to the coarse-grained reward cases, but it's unclear how the raw computational cost compares. The authors should present a comparison of the computational complexity of their approach against the coarse-grained baseline to clarify the feasability of their approach in practice.

2) The above weakness aside, the results in the 3 considered datasets are fairly good, although not definitively better than the baselines compared against across all metrics. E.g., generation metrics appear to be better for Quark and PPLM in Table 1 for sentiment analysis. Similarly, fluency and diversity + human eval appears to be leaning towards Quark in Table 2 for toxicity datasets.

3) The authors present convergence speed results in Figure 2 for one of the datasets, they should do the same for the other 2 datasets as well.

4) The objective function (9) being optimized here is a linear combination of likelihood for high rewarding tokens, unlikelihood for low rewards, entropy regularization for output diversity, and KL regularization for consistency with the anchor policy. However, the sensitivity of the results to the weights of these objective function components is unclear, as the authors use a distinct static set of weights for these components (mentioned in Section C.4). What is the overall sensitivity of the method to these parameters?

Minor nits/typos:
1) [Equation 2] Should there not be an expectation/sum over s' as well in this expression?
2) [Page 2] "which can be considered that action feedbacks from the same sentence are equal," did not understand what this is conveying. Are you saying that the use of a sentence-level reward function implies that the reward is equal for any valid subsequence of a given sentence?
3) [Page 3] "f (y, c) is a [pratical] classifier rating" pratical -> practical
4) Equation 16 from the appendix should be moved up to the main paper, where references are made to it.
5) [Page 8] "Results and Analysis. As shown in Tabel 3" Tabel -> Table

**Questions:**

1) What is the cost of computing/collecting fine grained feedback?

2) How sensitive are the results to the choice of objective function weightings in (9)?

---

### Official Review · Reviewer_JiMb · 2023-10-28

**Soundness:** 2 fair
**Presentation:** 2 fair
**Contribution:** 2 fair
**Rating:** 3
**Confidence:** 4

**Summary:**

In this paper, the authors introduce a reinforced learning algorithm with FIne-grained REward named FIRE. This FIRE with the reward function and find the trade-off between reward and training stability. Experiments on text generation with sentiment control, detoxification, and unlearning repetition are shown in the paper.

**Strengths:**

1. Controllable text generation with a fine reward is an exciting direction to explore.
2. The authors try to construct a reward function to better describe the reward during the learning while still preserving the training stability.

**Weaknesses:**

1. This paper conducted several experiments. However, I don't think the baselines the paper compares with are sufficient. Several works focus on a similar idea about incorporating the reward into text learning, such as RLPrompt [1] and AutoPrompt [2]. Those should become the baselines to compare the method proposed in the paper. Also, For controllable text generation, there is an interesting direction to utilize the diffusion process, such as the Diffusion-LM [3]. However, none of these are included and compared in the paper. Thus, I am not convinced with the experimental results shown in the paper.

2. The performance of the proposed method does not show enough improvements compared to the baseline mentioned in the paper. Could authors elaborate on what causes those falling behind?

3. The training stability is an important motivation of the paper. However, in the results section, I don't see enough evidence of the training stability. For example, what does the potential training loss look like compared to all baselines?

4. For the ablation section, what would be the efficiency comparison between the proposed method and the baselines? Such as the running time and computation latency.


[1] Rlprompt: Optimizing discrete text prompts with reinforcement learning

[2] AutoPrompt: Eliciting Knowledge from Language Models with Automatically Generated Prompts

[3] Diffusion-LM Improves Controllable Text Generation

**Questions:**

Please refer to the Weakness section.

---

### Official Review · Reviewer_EvSr · 2023-11-01

**Soundness:** 2 fair
**Presentation:** 3 good
**Contribution:** 2 fair
**Rating:** 5
**Confidence:** 2

**Summary:**

This work aims to improve the controllability of text generation in large language models by introducing a fine-grained reward mechanism in reinforcement learning. Current RL for LM techniques use sentence-level or paragraph-level feedback, which require more learning steps and larger exploration scale, resulting in limited controllability and slower convergence. The authors propose a novel reinforcement learning algorithm called FIRE, which utilizes a token-level reward function. FIRE achieves superior controllability and better performance with fewer learning steps compared to existing reinforcement learning methods.

**Strengths:**

The results show that FIRE attains improved performance for three tasks: sentiment-controlled text generation, detoxification, and unlearning repetition

**Weaknesses:**

- The introduction lacks precision: it does not explain 1) how bayesian factorization is used 2) how this method is trained 3) what does it mean to "avoid specific reward value"
- Much of paper describes this method as an alternative for policy gradient, however the experiments only show empirical evidence from fine-tuning language models. The authors should constrain scoping accordingly
- Empirical results do not show statistical significance, and the degree of improvement upon prior work is debatable

**Questions:**

- In table 1, some of the highlighted numbers are no the best numbers - what does this mean?
- Equation 5 (approximation) implies an existence of some function f that can assign utility to partial completion. If this is the case, then why not use f(y_{k+1}, c) - f(y_k, c) for credit assignment instead of the much more complicated proposal? At the very least, it seems like an important baseline comparison.